# Comparison of the disinfecting effect of sodium hypochlorite aqueous solution and surfactant on hospital kitchen hygiene using adenosine triphosphate swab testing

Takashi Aoyama[1,2]*, Tomoko Kudo[1,2]

1 Dietary Department, Shizuoka Cancer Centre, Shizuoka, Japan, 2 Division of Infection Control Nurse, Shizuoka Cancer Center, Shizuoka, Japan

* t.aoyama@scchr.jp

**Data Availability Statement:** The data underlying this study are available as Supporting information and at https://figshare.com/articles/dataset/ATP_RLU_data_xls/14123576.

## Abstract

The Japanese Ministry of Health requires large-scale cooking facilities to use sodium hypochlorite aqueous solution (HYP) on food preparation tools, equipment, and facilities to prevent secondary contamination. This study aimed to compare the disinfecting effect of HYP and surfactant using adenosine triphosphate (ATP) swab testing on large-scale equipment and facilities that could not be disassembled and disinfected in hospital kitchen. From May 2018 to July 2018, ATP swab tests were performed on the following six locations in the Shizuoka Cancer Centre Dietary Department Kitchen: cooking counter, mobile cooking counter, refrigerator handle, conveyor belt, tap handle, and sink. Six relative light unit (RLU) measurements were taken from each location. The $\log_{10}$ values of the RLU measurements were evaluated by dividing the samples into two groups: the control group (surfactant followed by HYP swabbing) and the HYP group (HYP swabbing only). The results showed that the RLUs ($\log_{10}$ values) in both the groups improved after disinfection ($p < 0.05$), except for the RLUs ($\log_{10}$ values) of the mobile cooking counter, tap handle, and sink in the control group after the HYP swab. The changes in the RLU ($\log_{10}$ value) did not differ between the two groups for all locations of the kitchen. Hence, HYP swabbing of large-scale equipment and facilities provides the same level of disinfection as surfactants and can be as beneficial.

## Introduction

Effective disinfection against norovirus infection is difficult to achieve because it is highly resistant to many common disinfection protocols [1]. The Japanese Ministry of Health, Labour and Welfare's "Hygienic Management Manual for Large-Scale Cooking Facilities" applies to cooking facilities that provide 300 or more meals at a time or 750 or more meals in a day [2]. These measures comply with the Hazard Analysis and Critical Control Points (HACCP) principles and state that food preparation tools (e.g., cutting boards) should be soaked in sodium hypochlorite aqueous solution (200 ppm) (HYP) [3], or when soaking is difficult, the same

**Funding:** The author(s) received no specific funding for this work.

**Competing interests:** NO authors have competing interests.

agent should be used to wipe the equipment and facilities to prevent secondary contamination by microbes such as norovirus [4, 5]. Large immovable equipment is easier to disinfect with HYP. Since the use of surfactants requires water, it also requires additional labor to maintain a dry floor surface. Therefore, the Shizuoka Cancer Centre Dietary Department Kitchen (SCC Kitchen) applies the same disinfecting agent to large equipment and facilities that cannot be disassembled and disinfected. While the manual does not require the testing of the hygiene status of large equipment and facilities, the SCC Dietary Department has introduced the use of relative light unit (RLU) values measured by adenosine triphosphate (ATP) swab testing [6], which has been shown to correlate with the colony-forming unit values of pre-disinfecting microbial cultures [7]. It has been reported that ATP swab testing should be conducted in conjunction with the colony-forming unit values [8]. Previous reports have suggested that ATP does not react with HYP [9] and that the hypochlorite ion ($ClO^-$) in HYP has a disinfecting effect [10]. In light of these reports, it was hypothesized that the disinfecting effect of HYP and surfactants on large equipment and facilities in hospital kitchens could be compared using ATP swab tests. However, there are several disadvantages associated with the use of HYP such as: (a) contamination with foreign bodies (organic substances) which quickly deactivates HYP, (b) HYP can decompose depending on the storage temperature and light exposure, (c) it cannot be prepared in advance, (d) it causes roughening of workers' hands and has other handling issues such as the effort required in preparing diluted solutions, (e) unstable effects such as metal corrosion, and (f) adverse effects on the respiratory tract [11]. In addition, no cleaning effect has been reported for an aqueous solution of sodium hypochlorite; however, it has been shown to have a disinfecting effect on the severe acute respiratory syndrome coronavirus 2 (SARS-CoV-2), which is responsible for the COVID-19 disease [12, 13]. The purpose of this study was to compare the disinfecting effect of HYP and surfactant on large equipment and facilities using ATP swab testing and to explore the benefits of HYP. It was hypothesized that the disinfecting effect of HYP swabbing on large-scale cooking facilities may outweigh its operational disadvantages, as long as it is properly handled.

## Materials and methods

### Examination of the disinfecting effect of HYP

ATP swab tests were conducted six times on random days between May 2018 and July 2018. The SCC Dietary Department had a floor area of 712.05 $m^2$ (including rest rooms), had six hospital staff and approximately 40 contracted staff, and served approximately 1,000 hospital meals. This study examined six locations (refrigerator handle; conveyor belt, having electrical system; cooking counter; mobile cooking counter; tap handle; and sink) with which food material and cooking staff came into contact most often, and thus, making the maintenance of strict hygiene of these locations necessary.

This study did not involve human subjects, and therefore, ethics board approval and informed consent were not required. The experimental procedures using the RLU measurements from ATP swab testing are shown in Fig 1. The disinfecting effect was examined by dividing the samples into two groups: the control group, wherein the samples were disinfected with the surfactant swab (70%; 10-fold dilution: pH 8.0, Advantec® Test Paper: Advantech Technologies Japan Corp) followed by the HYP (200 ppm; pH: 9.4, Advantec® Test Paper) swab; and the HYP group, wherein the samples were disinfected with the HYP (200 ppm; pH: 9.4, Advantec® Test Paper) swab only [14, 15].

ATP swab tests were performed at the end of the lunchtime meal distribution using conveyor belts (11:30 AM to 12:00 noon), with each of the six locations being tested a total of six times to obtain the RLU values: pre-disinfecting (A), post-surfactant swab (B), and post-HYP

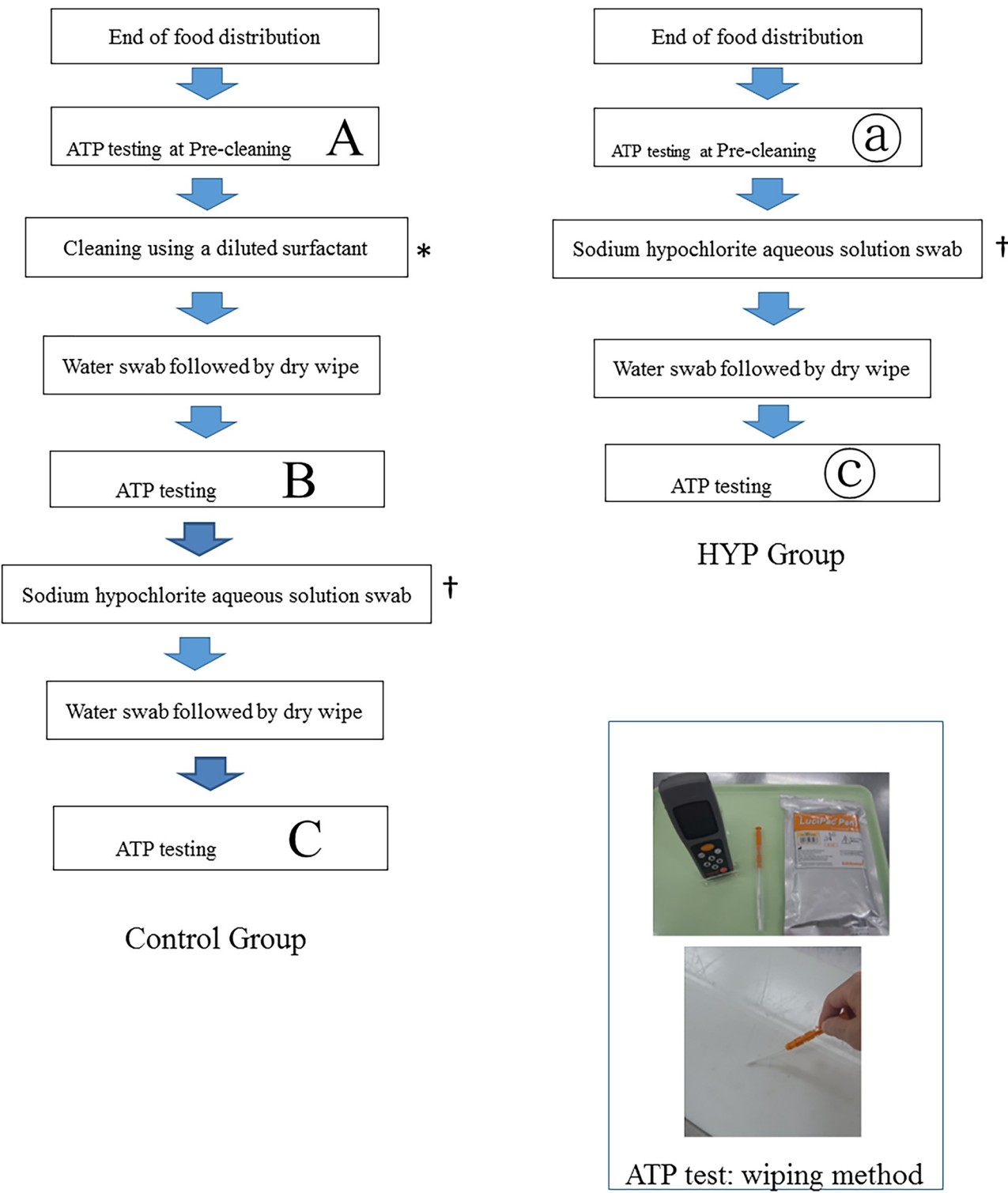

**Fig 1. Evaluation of cleaning/disinfection by ATP testing.** Experimental procedures for cleaning and disinfecting large equipment and facilities that could not be disassembled and disinfected using adenosine triphosphate (ATP) testing in the Shizuoka Cancer Centre Dietary Department Kitchen.

swab (C) for the control group, and pre-disinfecting (ⓐ) and post-HYP swab (ⓒ) for the HYP group; this was done to evaluate the RLU values and $log_{10}$ values (control group: three specimens × six locations × six replicates; HYP group: two specimens × six locations × six replicates). The changes in the $log_{10}$ of RLU values from the pre-disinfecting (A) to post-surfactant swab (B) for the control group, and from the pre-disinfecting (ⓐ) to post-HYP swab (ⓒ) for the HYP group were compared. In addition, the changes in $log_{10}$ of RLU values from the pre-disinfecting (A) to post-HYP swab (C) for the control group and from the pre-disinfecting (ⓐ) to post-HYP swab (ⓒ) for the HYP group were compared. Lumitester®PD-20 (Kikkoman Biochemifa Company, Tokyo, Japan) was used for ATP swab testing. To perform the tests, the RLU value of each test location was measured by swabbing a 10 cm² area of the test location (cooking counter: 10 cm² of all four counters; mobile cooking counter: 10 cm² of 15 counters; refrigerator handle: back and forth and front and back of eight handles; conveyor belt: 10 cm² of 20 m; tap handle: back and forth of eight handles; sinks: 10 cm² of the side walls of four sinks) with a LuciPac®Pen soaked in tap water (residual chlorine concentration: 0.2 ppm, determined by a residual chlorine measuring reagent), in accordance with the method of measurement described in the manufacturer's product specifications. When the RLU value was zero, the measurement was re-taken as it might not have been taken correctly initially.

## Statistical analysis

The evaluated items are presented as median values (minimum–maximum). Normality was evaluated using the Shapiro–Wilk test. The RLU values were converted to a common logarithm ($log_{10}$) to obtain normality, and the changes in measurements before and after disinfecting in the control group and HYP group as well as the differences in changes between the two groups were evaluated using the paired Student's t-test. The statistical analyses were done using JMP® 12 (SAS Institute Inc., Cary, NC, USA). The results of statistical analyses are shown by 95% confidence intervals. The results with $p$ value less than 0.05 were considered statistically significant.

## Results

The six test locations were tested six times each after the lunchtime meal distribution for the control group and HYP group by ATP swab testing (on May 23, 2018; June 6, 11, 18, and 24, 2018; and July 24, 2018). The median number of meals (range: minimum–maximum) was 367 (344–385; $p = 0.83$), and the temperature and humidity in the kitchen were 22.5˚C (20.8–24.1˚C; $p = 0.86$) and 66.0% (61.0–74.0%; $p = 0.70$), respectively [16], which meant that experimental conditions related to the degree of contamination, including the number of meals provided, were in place. Table 1 shows the RLU and $log_{10}$ values of the six test locations measured by ATP swab testing.

Table 2 and Figs 2 to 4 show the comparison of RLU and $log_{10}$ values in the six test locations measured by ATP testing. In all the six test locations, there were significant improvements in the $log_{10}$ values of the RLUs from pre-disinfecting (A) to post-surfactant swab (B) and post-HYP swab (C) for the control group and from pre-disinfecting (ⓐ) to post-HYP swab (ⓒ) for the HYP group ($p<0.05$). There was no significant difference in the $log_{10}$ values of the RLUs between post-surfactant swab (B) and post-HYP swab (C) for the control group at the mobile cooking counter, tap handle, and sink ($p = 0.52$, Fig 2 bottom; $p = 0.92$, Fig 4 top; $p = 0.17$, Fig 4 bottom); however, differences were observed at these time points for the control group at the cooking counter, refrigerator handle, and conveyor belt ($p<0.05$). In all the six test locations, there were no significant differences in the changes in $log_{10}$ values of the RLUs between post-

**Table 1. Results of RLU values and $\log_{10}$ values determined by ATP testing (n = 6).**

| | | | Median | Minimum | Maximum | $p^a$ |
|---|---|---|---|---|---|---|
| **Cooking counter** | **RLU value** | A | 15271 | 8277 | 58227 | 0.10 |
| | | B | 435 | 38 | 1191 | 0.29 |
| | | C | 75 | 31 | 304 | 0.03 |
| | | ⓐ | 37565 | 5539 | 115525 | 0.38 |
| | | ⓒ | 283 | 17 | 916 | 0.10 |
| | **$\log_{10}$ value** | A | 4.1 | 3.9 | 4.8 | 0.15 |
| | | B | 2.5 | 1.6 | 3.1 | 0.44 |
| | | C | 1.9 | 1.5 | 2.5 | 0.60 |
| | | ⓐ | 4.6 | 3.7 | 5.1 | 0.26 |
| | | ⓒ | 2.2 | 1.2 | 3.0 | 0.23 |
| **Mobile cooking counter** | **RLU value** | A | 29250 | 2346 | 514254 | <0.05 |
| | | B | 162 | 36 | 576 | 0.16 |
| | | C | 88 | 27 | 214 | 0.02 |
| | | ⓐ | 41513 | 2744 | 100474 | 0.75 |
| | | ⓒ | 200 | 61 | 2471 | 0.01 |
| | **$\log_{10}$ value** | A | 4.4 | 3.4 | 5.7 | 0.86 |
| | | B | 2.1 | 1.6 | 2.8 | 0.14 |
| | | C | 1.9 | 1.4 | 2.3 | 0.22 |
| | | ⓐ | 4.6 | 3.4 | 5.0 | 0.12 |
| | | ⓒ | 2.3 | 1.8 | 3.4 | 0.55 |
| **Refrigerator handle** | **RLU value** | A | 4602 | 808 | 10358 | 0.55 |
| | | B | 158 | 95 | 769 | 0.04 |
| | | C | 42 | 14 | 378 | 0.01 |
| | | ⓐ | 7921 | 2172 | 19092 | 0.75 |
| | | ⓒ | 208 | 156 | 585 | 0.03 |
| | **$\log_{10}$ value** | A | 3.6 | 2.9 | 4.0 | 0.64 |
| | | B | 2.2 | 2.0 | 2.9 | 0.18 |
| | | C | 1.6 | 1.1 | 2.6 | 0.55 |
| | | ⓐ | 3.9 | 3.3 | 4.3 | 0.89 |
| | | ⓒ | 2.3 | 2.2 | 2.8 | 0.08 |
| **Conveyor belt** | **RLU value** | A | 22938 | 8509 | 53190 | 0.49 |
| | | B | 932 | 394 | 2791 | 0.21 |
| | | C | 256 | 40 | 438 | 0.76 |
| | | ⓐ | 21568 | 11079 | 46128 | 0.23 |
| | | ⓒ | 391 | 189 | 5257 | <0.05 |
| | **$\log_{10}$ value** | A | 4.4 | 3.9 | 4.7 | 0.53 |
| | | B | 3.0 | 2.6 | 3.4 | 0.85 |
| | | C | 2.4 | 1.6 | 2.6 | 0.12 |
| | | ⓐ | 4.3 | 4.0 | 4.7 | 0.58 |
| | | ⓒ | 2.6 | 2.3 | 3.7 | 0.15 |

(*Continued*)

**Table 1.** (Continued)

| | | | Median | Minimum | Maximum | $p^a$ |
|---|---|---|---|---|---|---|
| **Tap handle** | **RLU value** | A | 10329 | 7868 | 29630 | 0.05 |
| | | B | 136 | 61 | 371 | 0.29 |
| | | C | 97 | 38 | 561 | 0.01 |
| | | ⓐ | 18104 | 5310 | 44148 | 0.43 |
| | | ⓒ | 162 | 64 | 1574 | <0.05 |
| | **log₁₀ value** | A | 4.0 | 3.9 | 4.5 | 0.23 |
| | | B | 2.1 | 1.8 | 2.6 | 0.55 |
| | | C | 2.0 | 1.6 | 2.7 | 0.16 |
| | | ⓐ | 4.2 | 3.7 | 4.6 | 0.84 |
| | | ⓒ | 2.2 | 1.8 | 3.2 | 0.34 |
| **Interior of sink** | **RLU value** | A | 2946 | 921 | 16134 | 0.08 |
| | | B | 471 | 192 | 1075 | 0.52 |
| | | C | 180 | 94 | 554 | 0.03 |
| | | ⓐ | 11004 | 2278 | 61966 | 0.02 |
| | | ⓒ | 639 | 263 | 975 | 0.57 |
| | **log₁₀ value** | A | 3.4 | 3.0 | 4.2 | 0.87 |
| | | B | 2.7 | 2.3 | 3.0 | 0.79 |
| | | C | 2.3 | 2.0 | 2.7 | 0.16 |
| | | ⓐ | 3.9 | 3.4 | 4.8 | 0.07 |
| | | ⓒ | 2.8 | 2.4 | 3.0 | 0.49 |

RLU, relative light unit; ATP, adenosine triphosphate

[a] $p$ values were calculated by the Shapiro–Wilk test.

**Table 2. Results of the changes in the RLUs (log₁₀ values) by ATP testing (n = 6).**

| | | | Median | Minimum | Maximum |
|---|---|---|---|---|---|
| **Cooking counter** | **Change in log₁₀ value** | A→B | -1.8 | -3.1 | -0.9 |
| | | A→C | -2.5 | -3.3 | -1.4 |
| | | ⓐ→ⓒ | -2.4 | -3.8 | -0.8 |
| **Mobile cooking counter** | **Change in log₁₀ value** | A→B | -2.1 | -3.3 | -1.8 |
| | | A→C | -2.6 | -3.4 | -1.54 |
| | | ⓐ→ⓒ | -1.9 | -3.2 | -1.4 |
| **Refrigerator handle** | **Change in log₁₀ value** | A→B | -1.5 | -1.9 | -0.02 |
| | | A→C | -2.1 | -2.7 | -0.3 |
| | | ⓐ→ⓒ | -1.6 | -2.0 | -0.6 |
| **Conveyor belt** | **Change in log₁₀ value** | A→B | -1.3 | -2.0 | -0.5 |
| | | A→C | -2.0 | -2.9 | -1.5 |
| | | ⓐ→ⓒ | -1.7 | -2.2 | -0.3 |
| **Tap handle** | **Change in log₁₀ value** | A→B | -1.9 | -2.7 | -1.4 |
| | | A→C | -2.0 | -2.3 | -1.2 |
| | | ⓐ→ⓒ | -2.0 | -2.5 | -1.2 |
| **Interior of sink** | **Change in log₁₀ value** | A→B | -0.9 | -1.7 | -0.1 |
| | | A→C | -1.1 | -2.2 | -0.5 |
| | | ⓐ→ⓒ | -1.3 | -2.1 | -0.4 |

RLU, relative light unit; ATP, adenosine triphosphate

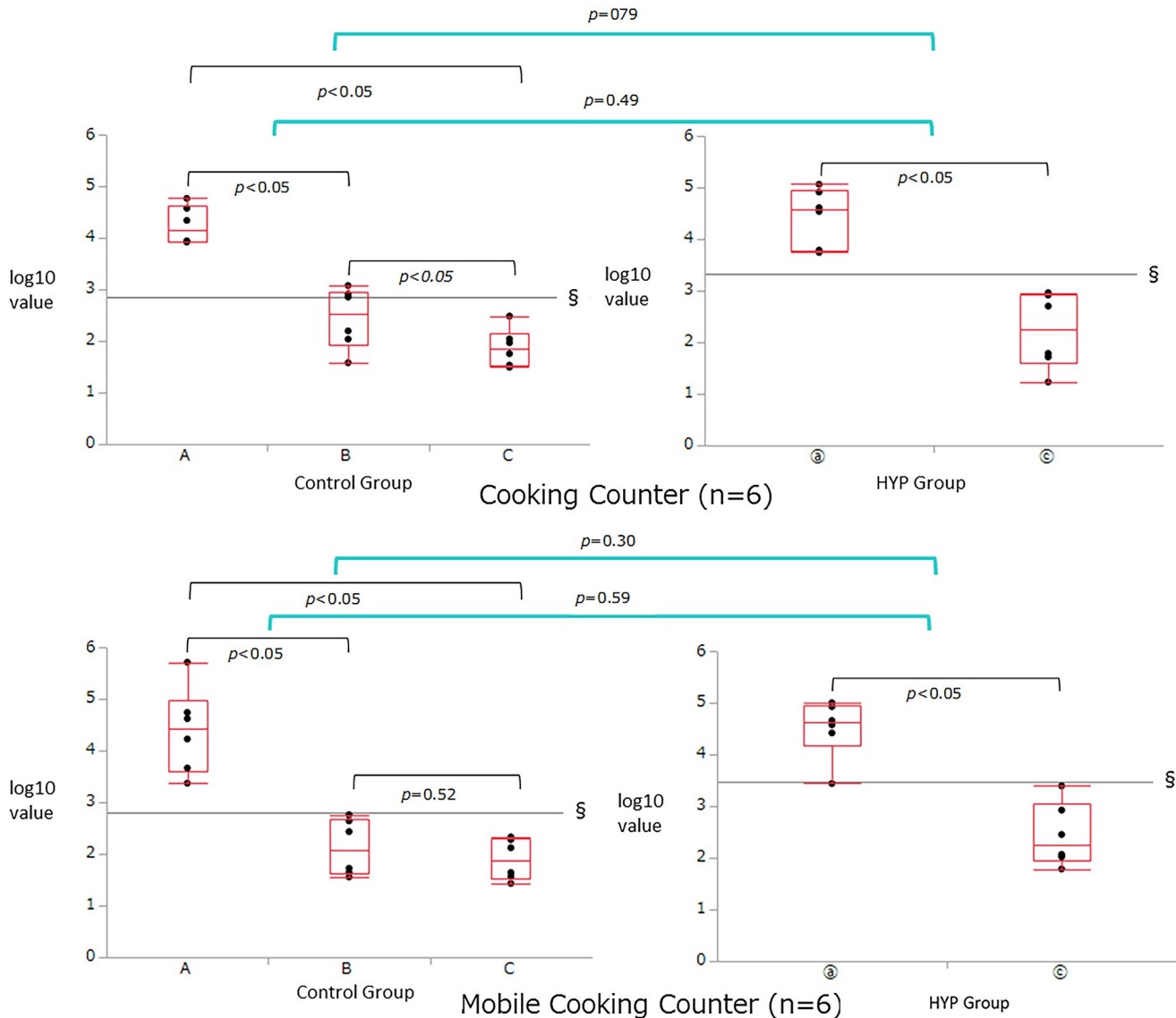

**Fig 2. Inter-group comparison of the relative light unit (RLU) and log$_{10}$ values in the cooking counter and mobile cooking counter test locations measured by adenosine triphosphate (ATP) swab testing.** There were significant improvements in the log$_{10}$ values of the RLUs from the pre-disinfecting (A) to post-surfactant swab (B) and post-HYP swab (C) for the control group, and from the pre-disinfecting (ⓐ) to post-HYP swab (ⓒ) for the HYP group ($p<0.05$). There was no significant difference in the log$_{10}$ value of RLU values between the post-surfactant swab (B) and post-HYP swab (C) for the control group at the mobile cooking counter. ($p = 0.52$); however, differences were observed at these time points for the control group at the cooking counter. In both these locations, there were no significant differences in the changes in the logarithmic RLU values between the post-HYP swab (ⓒ) of the HYP group and the post-surfactant swab (B) and post-HYP swab (C) of the control group.

HYP swab (ⓒ) of the HYP group and the post-surfactant swab (B) and post-HYP swab (C) of the control group (Figs 2 to 4).

## Discussion

While the Hygienic Management Manual for Large-Scale Cooking Facilities [2], which follows the HACCP, does not mandate the testing of environmental contamination, this study showed that the pre-disinfecting RLU values of large equipment and facilities in kitchens were as high as those that have been previously reported in other studies [7, 8], thus making disinfection

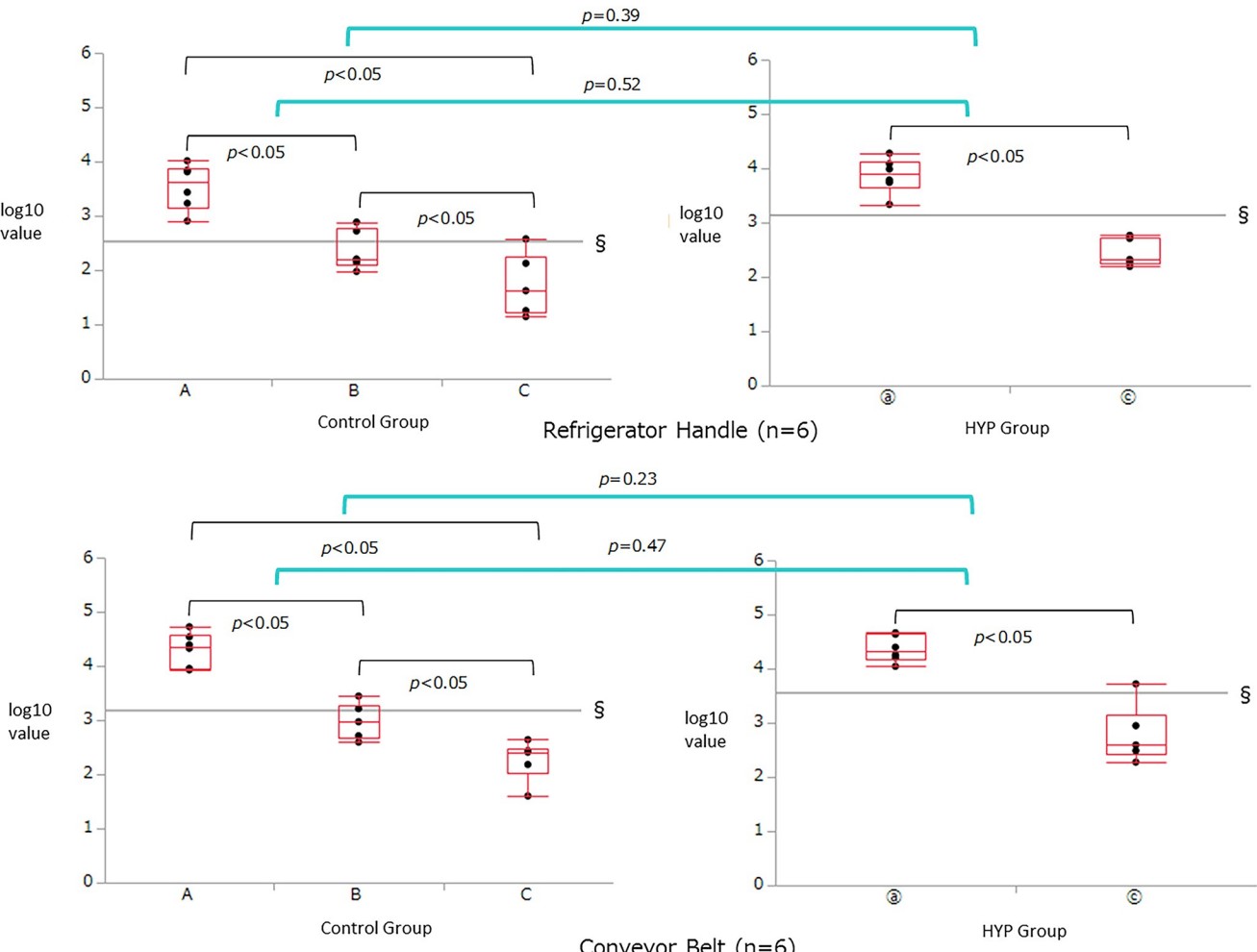

**Fig 3. Inter-group comparison of the relative light unit (RLU) and log$_{10}$ values in the refrigerator handle and conveyor belt test locations measured by adenosine triphosphate (ATP) swab testing.** There were significant improvements in the log$_{10}$ values of the RLUs from the pre-disinfecting (A) to post-surfactant swab (B) and the post-HYP swab (C) for the control group, and from the pre-disinfecting (ⓐ) to post-HYP swab (ⓒ) for the HYP group ($p < 0.05$). Significant differences in the log$_{10}$ values of the RLUs between the post-surfactant swab (B) and post-HYP swab (C) for the control group were observed at the refrigerator handle and conveyor belt ($p < 0.05$). In both these locations, there were no significant differences in the change in log$_{10}$ values of the RLUs between the post-HYP swab (ⓒ) of the HYP group and the post-surfactant swab (B) and post-HYP swab (C) of the control group.

necessary (Table 1: A, ⓐ). Apart from the post-surfactant swab (B) and post-HYP swab (C) of the mobile cooking counter, tap handle, and sink for the control group, all the log$_{10}$ values of RLUs showed significant improvement, and it was shown that the disinfecting effect could be further improved by HYP swabbing for the cooking counter, refrigerator handle, and conveyor belt. There were no significant differences in the changes in the log$_{10}$ values for both the groups. These results suggested that HYP swabbing for large equipment and facilities may have a disinfecting effect that is equivalent to that achieved by the surfactant. However, due to the instability and operational disadvantages associated with HYP (10%) in the SCC Kitchen, it is stored at 18˚C and away from light before diluting to 200 ppm for use and requires handling with disposable gloves. This study did not find any adverse events (respiratory problems, dermatological irritation, or metal corrosion) among cooking staff or large equipment when this agent was used, and such observations have not been reported by the National Consumer

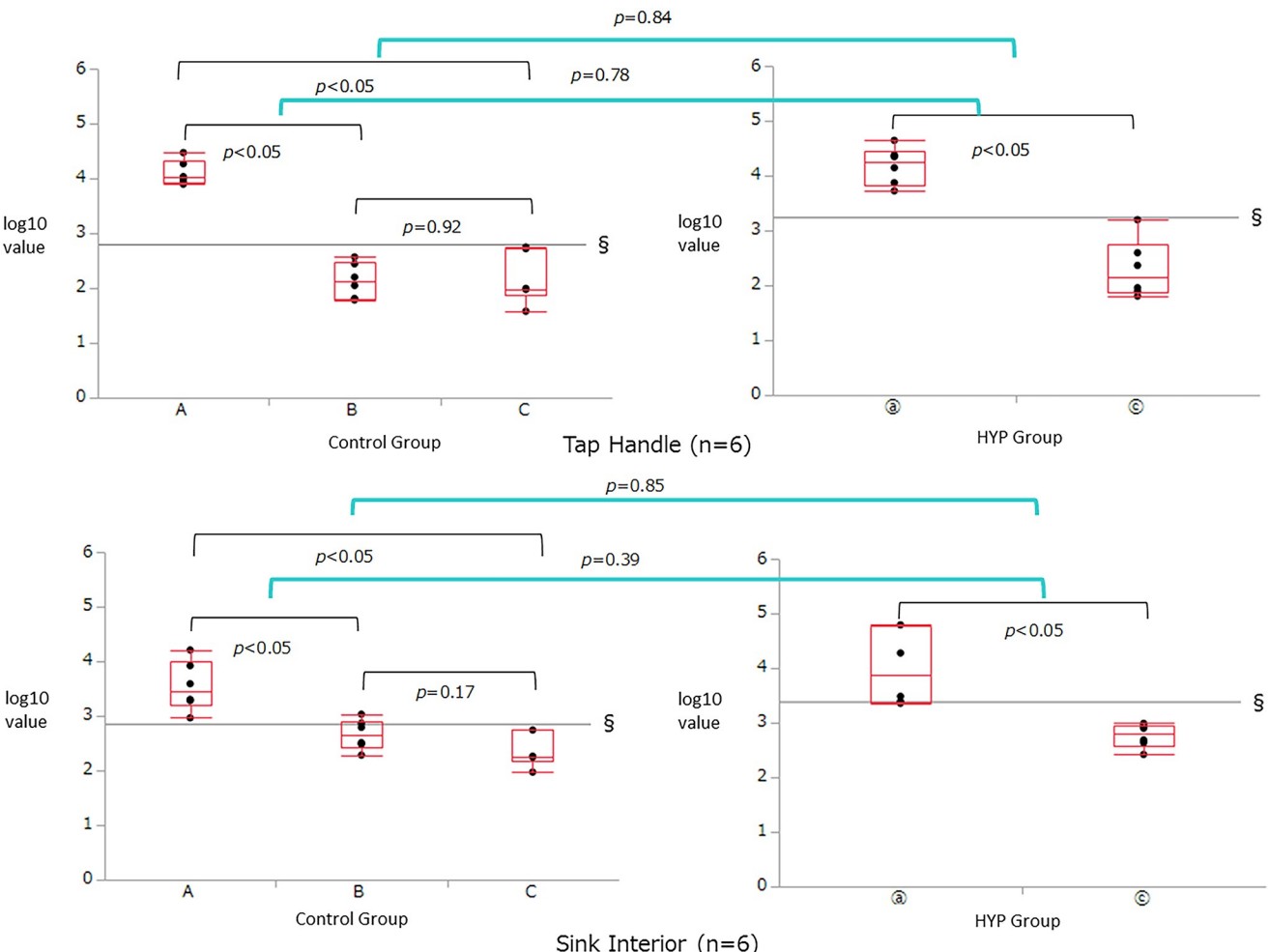

**Fig 4. Inter-group comparison of the relative light unit (RLU) and log$_{10}$ values in the tap handle and sink test locations measured by adenosine triphosphate (ATP) swab testing.** There were significant improvements in the log$_{10}$ values of the RLUs from the pre-disinfecting (A) to post-surfactant swab (B) and the post-HYP swab (C) for the control group, and from the pre-disinfecting (ⓐ) to post-HYP swab (ⓒ) for the HYP group ($p<0.05$). There were no significant differences in the log$_{10}$ values of the RLUs between the post-surfactant swab (B) and post-HYP swab (C) for the control group at the tap handle and sink ($p = 0.92$, Fig 4 top; $p = 0.17$, Fig 4 bottom). In both these locations, there were no significant differences in the change in log$_{10}$ values of the RLUs between the post-HYP swab (ⓒ) of the HYP group and the post-surfactant swab (B) and post-HYP swab (C) of the control group.

Affairs Centre [11, 12]. Hence, this study showed that the advantages of the disinfecting effect of HYP swabbing outweigh its operational disadvantages, and that there may be benefits associated with using this agent.

Noroviruses are vulnerable to heating at >85˚C, but it is difficult to use boiling water because large equipment and conveyor belts have electrical systems [4, 17, 18]. For this reason, disinfection methods other than HYP have been reported. Alcohol preparations may or may not have a disinfecting effect on viruses, therefore alcohols (especially, 1-propanol) are as effective as low concentrations (about 100–200 ppm) of sodium hypochlorite, so sprays are used to disinfect the clean facility environment [19–21]. Although it has been reported that alcohols are effective against ultraviolet rays, gamma rays, and microbubbles, they are not suitable for large equipment and facilities [22–25]. Owing to the aforementioned reports, only HYP is recommended for norovirus in Japan [4]. The aqueous solution of sodium hypochlorite has also

been shown to be effective against enveloped viruses such as the SARS-CoV-2 [13, 26]. The findings of the present study, which demonstrated the potential of HYP for general cleaning purposes, were considered to be significant. However, in this study, no other medium method was employed along with ATP swabbing [7, 8, 27]. The ATP approach is unable to detect viruses and its findings should be critically evaluated to elucidate whether the ATP endpoint for a bioassay would be appropriate to monitor microbes as well as pathogenic molecules like viruses [28–31]. There is also a room for consideration as to whether the fungi can also be detected by ATP [32]. In addition, the involvement of ATP and basicity (pH) remains unclear [33, 34].

This study has some limitations. First, it did not compare between surfactant and HYP in terms of resource costs and reduction in the amount of work required. Second, this study did not explore the possibility of avoiding the wastage of human and water resources required for washing and the pollution caused by the waste. However, it should be noted that there have been very few studies that explored these possibilities.

In conclusion, this study used ATP swab testing to show that HYP swabbing exhibited a disinfecting effect similar to that of surfactant for large equipment and facilities in the hospital kitchen that cannot be disassembled and disinfected. Therefore, HYP, which has better disinfecting properties, may be a suitable disinfectant for such equipment.

## Supporting information

**S1 Data.**
(XLS)

## Acknowledgments

I would like to thank Editage (www.editage.com) for English language editing.

## Author Contributions

**Conceptualization:** Takashi Aoyama.

**Data curation:** Takashi Aoyama.

**Formal analysis:** Takashi Aoyama.

**Funding acquisition:** Takashi Aoyama.

**Investigation:** Takashi Aoyama, Tomoko Kudo.

**Methodology:** Takashi Aoyama.

**Project administration:** Takashi Aoyama.

**Resources:** Takashi Aoyama.

**Software:** Takashi Aoyama.

**Supervision:** Takashi Aoyama.

**Validation:** Takashi Aoyama.

**Visualization:** Takashi Aoyama.

**Writing – original draft:** Takashi Aoyama.

**Writing – review & editing:** Takashi Aoyama.

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
