## [Decision Letter · Decision Letter 0]

9 Feb 2021

PONE-D-20-31068

Comparison of the disinfecting effect of sodium hypochlorite aqueous solution and surfactant on hospital kitchen hygiene using adenosine triphosphate swab testing

PLOS ONE

Dear Dr. Takashi Aoyama,

Thank you for submitting your manuscript to PLOS ONE. After careful consideration, we feel that it has merit but does not fully meet PLOS ONE’s publication criteria as it currently stands. Therefore, we invite you to submit a revised version of the manuscript that addresses the points raised during the review process.

We look forward to receiving your revised manuscript.

Kind regards,

Hans-Uwe Dahms, Ph.D.

Academic Editor

PLOS ONE

Additional Editor Comments:

The work was conducted carefully and provides sufficient data. The flow of thought is convincing and the English is suitable.

Unfortunately, INTRODUCTION and DISCUSSION (of own RESULTS and those of others are not detailed at this point! It is suggested that the authors add more reflections around the many MORE disinfection methods (incl. NON-CHEMICAL), comparison with available protocols WORLDWIDE, and report more DISCUSSION about these issues. For example ATP- approach will NOT detecet VIRUSES and it should be critically evaluated whether the ATP endpoint for a bioassay is appropriate to monitor MICROBES as well as pathogenic molecules like viruses....Would fungi be equally assayed with ATP? A minimum of 25-30 REFERENCES would be necessary to provide an in depth DISCUSSION.

Journal Requirements:

2.) In your Methods section, please provide additional information regarding the permits you obtained for the work. Please ensure you have included the full name of the authority that approved the field site access and, if no permits were required, a brief statement explaining why.

3.) In your Data Availability statement, you have not specified where the minimal data set underlying the results described in your manuscript can be found. PLOS defines a study's minimal data set as the underlying data used to reach the conclusions drawn in the manuscript and any additional data required to replicate the reported study findings in their entirety. All PLOS journals require that the minimal data set be made fully available. For more information about our data policy, please see http://journals.plos.org/plosone/s/data-availability.

4.) Please amend your list of authors on the manuscript to ensure that each author is linked to an affiliation. Authors’ affiliations should reflect the institution where the work was done (if authors moved subsequently, you can also list the new affiliation stating “current affiliation:….” as necessary).

Reviewers' comments:

Reviewer's Responses to Questions

**Comments to the Author**

1. Is the manuscript technically sound, and do the data support the conclusions?

Reviewer #1: Partly

Reviewer #2: Yes

2. Has the statistical analysis been performed appropriately and rigorously? 

Reviewer #1: No

Reviewer #2: Yes

3. Have the authors made all data underlying the findings in their manuscript fully available?

Reviewer #1: Yes

Reviewer #2: No

4. Is the manuscript presented in an intelligible fashion and written in standard English?

Reviewer #1: Yes

Reviewer #2: Yes

5. Review Comments to the Author

Reviewer #1: After examining the manuscript, I am of the view that it is not suitable for publication in current form. Although the work appears to have been conducted carefully, unfortunately data, experiments and discussion are not enough, hence, not suitable for publication in this journal.

It is suggested that the authors should add more experimental data of various disinfection methods, comparison with available protocols, and report more discussion.

Reviewer #2: After examining the manuscript, I am of the view that it is not suitable for publication in current form, particularly since an IN DEPTH DISCUSSION and LITERATURE update is missing. The work was conducted carefully and provides sufficient data

for its objectives. The flow of thought is convincing and the English is suitable.

Unfortunately, INTRODUCTION and DISCUSSION (of own RESULTS and those of others are not enough as yet! It is suggested that the authors add more reflections around the many MORE disinfection methods (incl. NON-CHEMICAL), comparison with available protocols WORLDWIDE, and report more DISCUSSION about these issues.

For example ATP- approach will NOT detecet VIRUSES and it should be critically evaluated whether the ATP endpoint for a bioassay is appropriate to monitor MICROBES as well as pathogenic molecules like viruses....Would fungi be equally assayed with ATP?

A minimum of 25-30 REFERENCES would be necessary to provide an in depth DISCUSSION.

Are ALL THE DATA made available to the reader? If not, pls. provide those TABLE information as

Supplementary files.

I am adding an amended version of the MS herewith.

END

Hans-Uwe Dahms

6. PLOS authors have the option to publish the peer review history of their article (what does this mean?). If published, this will include your full peer review and any attached files.

Reviewer #1: No

Reviewer #2: No

---

## [Editor Report · Decision Letter 1]

25 Mar 2021

Comparison of the disinfecting effect of sodium hypochlorite aqueous solution and surfactant on hospital kitchen hygiene using adenosine triphosphate swab testing

PONE-D-20-31068R1

Dear Dr. Takashi Aoyama,

We’re pleased to inform you that your manuscript has been judged scientifically suitable for publication and will be formally accepted for publication once it meets all outstanding technical requirements.

Kind regards,

Hans-Uwe Dahms, Ph.D.

Academic Editor

PLOS ONE

Additional Editor Comments (optional):

This MS has now reached a state that makes it acceptable for publication.

---

## [Editor Report · Acceptance letter]

29 Mar 2021

PONE-D-20-31068R1 

Comparison of the disinfecting effect of sodium hypochlorite aqueous solution and surfactant on hospital kitchen hygiene using adenosine triphosphate swab testing 

Dear Dr. Aoyama:

I'm pleased to inform you that your manuscript has been deemed suitable for publication in PLOS ONE. Congratulations! Your manuscript is now with our production department. 

Kind regards, 

on behalf of

Dr. Hans-Uwe Dahms 

Academic Editor

PLOS ONE